# A value creation model from science-society interconnections: Archetypal analysis combining publications, survey and altmetric data

Irene Ramos-Vielba[1], Nicolas Robinson-Garcia[2]*, Richard Woolley[3]

**1** Danish Centre for Studies in Research and Research Policy, Department of Political Science, Aarhus University, Aarhus, Denmark, **2** EC3 Research Group, Information and Communication Studies Department, Universidad de Granada, Granada, Spain, **3** INGENIO (CSIC-UPV), Universitat Politècnica de València, Valencia, Spain

* elrobin@ugr.es

## Abstract

The interplay between science and society takes place through a wide range of intertwined relationships and mutual influences that shape each other and facilitate continuous knowledge flows. Stylised consequentialist perspectives on valuable knowledge moving from public science to society in linear and recursive pathways, whilst informative, cannot fully capture the broad spectrum of value creation possibilities. As an alternative we experiment with an approach that gathers together diverse science-society interconnections and reciprocal research-related knowledge processes that can generate valorisation. Our approach to value creation attempts to incorporate multiple facets, directions and dynamics in which constellations of scientific and societal actors generate value from research. The paper develops a conceptual model based on a set of nine value components derived from four key research-related knowledge processes: production, translation, communication, and utilization. The paper conducts an exploratory empirical study to investigate whether a set of archetypes can be discerned among these components that structure science-society interconnections. We explore how such archetypes vary between major scientific fields. Each archetype is overlaid on a research topic map, with our results showing the distinctive topic areas that correspond to different archetypes. The paper finishes by discussing the significance and limitations of our results and the potential of both our model and our empirical approach for further research.

## Introduction

Considerable scholarly and policy attention is devoted to the relationship between science and society, particularly the theme of societal returns on public investment in research. It has been argued that much of science is overly self-referential or oriented to economic returns at the expense of producing more diverse forms of 'public value' [1]. The rise of complex health,

**Data Availability Statement:** All relevant data used in the paper are openly accessible at https://doi.org/10.5281/zenodo.6393226 R script available at: https://github.com/elrobin/value-components

Codebook available at: https://rpubs.com/elrobin/value-components.

**Funding:** This research was supported by the Spanish Ministry of Economy, Industry and Competitiveness through the State Plan of Scientific and Technical Research and Innovation (EXTRA project, grant CSO2013-48053-R). RW was supported by the Oslo Institute for Research on the Impact of Science (OSIRIS, grant 256240) funded by the Research Council of Norway. NRG is currently supported by a Ramón y Cajal grant from the Spanish Ministry of Science (RYC2019-027886-I). The funders had no role in study design, data collection and analysis, decision to publish, or preparation of the manuscript.

**Competing interests:** The authors have declared that no competing interests exist.

environmental, and other problems, many at least partly as a consequence of scientific and technological development, has collapsed the artificial separation between scientists' pursuit of objective facts, and the highly contested world of values, culture, and problems [2]. It has been pointed out that scientific knowledge only has value in use [3, 4] and that the 'social robustness' of scientific knowledge is therefore of paramount importance [5]. The entwining of science and society is thus a subject of continuing vigorous contestation [6].

In an extensive body of work on the public value of science, Barry Bozeman and colleagues emphasized that the 'knowledge value collective' includes not just researchers, but also all those actors who use, or support the use of, research. According to Bozeman, '[t]he actual users. . .are the ones who, in practice, ascribe value', the effectiveness of a knowledge value collective 'will be related to its success in "marketing" its outputs' and supporting users to find them valuable [3]. In other words, the value of research is not intrinsic to knowledge outputs themselves but depends on the active conceptualisation and realisation of value by users. Numerous research impact evaluation processes have emerged that provide detailed frameworks for better understanding how research and users become connected [7], seeking to document the 'full pathway from research to impact, including knowledge exchange, outputs, outcomes, and interim impacts, to allow the route to impact to be traced' [8]. Research impact evaluation methods, such as the payback framework [9], ASIRPA [10], and SIAMPI [11] build sequential pathways at a project or programme level allied to a 'theory of change' leading toward documentable outcomes and structuring interpretations about predicted future impacts [12]. Such approaches are useful for reconstructing pathways between research activities, research outputs, and linked outcomes at the level of specific research projects, programmes, or organisations [10]. Each of these approaches treats the conceptualisation of societal value from research differently [13] and focuses on different configurations of connections between 'science and non-science' entities [14].

This paper seeks to contribute to this discussion and to empirical analysis of the generation of value from research in society, but in a different way to that found in the 'research impact' literature. Rather than trying to document a relatively narrow sequence of events that may lead to the attribution of a specific research impact, we focus on broad research-related knowledge processes that can contribute to generating societal value [3]. It is through the intertwined processes of knowledge production, translation, communication, and utilization that efforts to 'valorise' research by and for different research users occurs. Similar to Smit and Hessels [13], our conceptualisation of value from research is open, inclusive, and does not grant *a priori* ascendency to science or society. Based on this understanding, the paper conducts an exploratory empirical study of interconnections among research-related knowledge processes. The rationale for the paper is to investigate whether a 'grammar' or set of archetypal configurations can be discerned among these processes that structures science-society interconnections in particular ways. We also explore whether such configurations vary between different major scientific fields and how they map onto research topic areas. We see this method as complementary to existing approaches that trace forwards or backwards in order to follow the translation of research to society, or map the uptake of research outputs from knowledge reservoirs [15, 16].

In what follows, we develop a conceptual model for this objective and then conduct an empirical analysis, which we emphasize is exploratory research due to the acknowledged limitations of some of the available data proxies that we use. The article is structured as follows; we first review the major streams of literature on research-related knowledge processes that underpin the choice of components for our analytical model. Then, we describe our conceptual thinking and specify the components of our model derived from the previous section. In the next section, we describe our empirical method, the data and specific measures we use for

the various components of our model. The results of our archetype analysis are then presented, and we conclude by discussing the significance and limitations of our results and the potential of both our model and our empirical approach for further research.

## Value creation between science and society

A large but differentiated body of scholarly literature focuses on the institutions, organisations, actors, and activities that carry research-related knowledge processes. There are specialised literatures on how knowledge is produced, how it is translated into user contexts, how science is communicated, and how it is utilized in ways that provide new impetus to research and benefits for society. In this section we review, at a summary level, aspects of these bodies of knowledge with the main purpose of grounding the components of our conceptual model in existing scholarship.

### Research collaboration, translation, and engagement

Research collaboration lies at the heart of scientific practice and knowledge production [17]. Joint research involving public sector research organisations, including universities and non-academic partners, is a key mechanism by which the human capital and research infrastructures in public institutions are used for industrially oriented research driving new knowledge and technological progress [18–20]. Processes to co-produce knowledge [21] involving a range of non-academic partners [22] institutionalise modes of formal and informal engagement and interaction that can underpin durable collective research agendas, organisational forms, and innovation pathways [23–25]. A variety of intermediaries, including legal and market professionals, support the transfer of university knowledge and technology to external users and innovators, professionalising these innovation pathways [26–28]. At the heart of such networked processes lie interpersonal and inter-organisational relationships, the 'productive interactions' among researchers and stakeholders through which transformative learning occurs and new valorisation possibilities can emerge [11, 29].

The vast literature on joint and engaged research and technology transfer highlights an enormous range of individual, organisational and contextual factors [30] that contribute to shaping research-related knowledge processes. Several systemic accounts have historicised the evolution of interactions between scientific and societal agents, shaping knowledge production and translation [31]. 'Post-industrial science' and 'academic capitalism' perspectives both highlight the effects of competition for scarce public resources and the expansion of research in private or hybrid contexts and pointing out the reflexive influence of science and technology policy [32]. Academic capitalism notes the need for more entrepreneurial attitudes and practices on the side of public sector researchers under such conditions of scarcity [33]. Mode-2 knowledge production [34] describes a more socially distributed and context-dependent organising of knowledge production and evaluation leading to 'socially robust knowledge', the validity of which rests in a broad community of producers, disseminators and users of knowledge [5].

Building on national innovation systems theory, which privileges engagement between actors across institutional sectors [35], the triple-helix model of interdependent university, government, and industry spheres [36] mobilises hybrid mediating entities (including policy actors) as facilitators of knowledge activities among diverse sets of stakeholders [37], driving the concomitant transformation of public sector organisations into more engaged and entrepreneurial actors [38]. In addition, the perspective of the 'media-based and culture-based public' is integrated as a fourth helix in knowledge and innovation ecosystems [39]. The public and civil society play a significant role–through culture and values–in shaping a 'public reality',

often expressed via the media, that influences research processes and becomes essential for public support to science. Finally, post-normal science emphasises that citizens' participation in scientific research and evaluation is required to improve the relevance and legitimacy of science and technology. Crucially, following the dissolution of the strict demarcation between facts and values, scientific research and results need to be communicated and debated with more inclusive 'peer communities' prior to inclusion in policy processes [2]. Each of these systemic approaches foregrounds different aspects of the multi-actor networks that mobilise resources and capabilities to generate value from research. Calls to invest in and expand collaborations between public sector research organisations and partners from industry, government and society seek to further align or optimise science-society value processes to address societal missions and combat global crises [40].

## Open science and innovation

An emerging transversal dynamic that is re-shaping science and innovation encompasses a diverse array of practices, processes, and infrastructures under the heading of 'openness'. Open science or open research cultures refer to a range of practices and institutional arrangements designed to make science more transparent, reproducible, and accessible [41]. Numerous elements fit under the umbrella of openness in research and innovation, including open workflows, open data, open access publications, open-source software, open code, pre-prints, open evaluation, and citizen science [42]. The UNESCO Recommendations on Open Science [43] also include open educational resources, open engagement of societal actors, and openness to diversity of knowledge. In Europe, a concerted policy-push by the European Commission, including investments in infrastructure such as the Open Science Cloud are grounded in the rationale that valorisation of science will be more rapid and more extensive under open science conditions. The philosophy of 'open innovation' in firms has also encouraged more porous organisation boundaries, invigorating innovation [44]. Thus, 'openness' is understood in an all-inclusive sense referring to those practices, policies, objects, and institutional arrangements designed to enhance the research-based value that can be generated by eliminating obstructions and barriers to participation in the production and use of scientific knowledge by scientific and societal actors of all types.

Openness also improves the communication of information about science and of research results [42]. Ensuring data, code, and other research outputs are findable, accessible, interoperable and re-usable (FAIR) can facilitate the take-up and generation of value from research by multiple users. Research actors, including researchers/groups, centres, institutes, faculties, universities, and scientific publishers, all engage in the dissemination of science-related information and research results to academic and non-academic audiences using a range of different channels and peer networks [45, 46]. Most research funding organisations and programmes, including the Framework Programmes of the European Commission, now require exploitation and dissemination plans that detail how research progress and results with be circulated among potential users. Dissemination tools include a wide range of analogue and digital media that can circulate research and research findings to enlarge the numbers of potential users of research [47]. Research actors may also conduct demonstrations, produce targeted presentations, write reports, or use visualisations that are tailored to the interests of specific research users or potential beneficiaries [48]. Intermediary actors, including industry consultants, policy think-tanks, community organisations, and knowledge exchange specialists, can also function as knowledge brokers in different contexts to support efforts to transmit valorisation opportunities to potential research users [49, 50].

## Knowledge communication and diffusion

Whereas research dissemination activities typically involve research actors, they may not be directly involved in the promotion of science and research results through the various channels of media communications. Media professionals working in 'traditional' broadcast or publishing media, such as television, newspapers, or magazines, convey 'news' or information in a relatively unidirectional manner. Journalists, editors, and directors, and other professionals and technicians produce their own forms of outputs using current (science) events or substantive content, designed for consumption by the public or targeting demographic or other subgroups. News and media platforms are therefore involved in amplifying the reach of science and research results throughout society [51]. While the stakes of the game [52] in the media field share some similarities with the stakes of the scientific field, such as novelty and primacy, the media field is principally organised around its own conceptions of quality, such as production values, and indicators of success, such as relative audience share. However, media and other social fields overlap in that the visibility of successful news or other presentations of scientific results can increase societal awareness and potentially lead to the emergence of new opportunities for valorisation of those results.

Social media are communication channels with distinctive properties that shape the proliferation of science-society interactions [14]. Actors of all kinds, including researchers and potential research users, government, and citizens, can communicate directly [53–55], and participate in heterogeneous information networks around special interests and issues [56, 57]. Real-time knowledge-focused social media interactions include monitoring and managing crisis events [58], and tracking and tracing research use for public health outcomes [59]. Costas and colleagues propose a general framework for social media structuring of science-society interactions as 'heterogeneous couplings' or the 'co-occurrence of science and non-science objects, actors, and interactions' [14]. From our perspective, of particular interest is the potential of social media to amplify science communication through mechanisms including 'likes' and 're-tweets' (Twitter) and 'shares' (Facebook). Information amplification can extend awareness and interest in science and research results. Research actors' intentional actions may be limited to simply seeding a social media platform with an information object (text, video, diagram, etc.), with subsequent amplification being a relatively independent and ungoverned process (although subject to professional information promotion practices).

## Knowledge utilization

Finally, the utilisation of research, research outputs, and scientific knowledge-based products and processes generates value across myriad scientific/societal contexts. Expert or proficient end-users of (primarily) technological artefacts produce feedback loops and user communities that interact with research and development focused actors [60]. Open-source software communities use research to construct new objects and processes in the innovation commons [61]. These may be partially translated back (or act as insurgents) into wider research and development and research user communities or networks, such as in the case of Linux and Microsoft [62], whilst partially remaining in the hands of open-source communities. Weiss [16, 63] observed how scientific knowledge could 'creep' into society, particularly through policy processes that reflect a process of 'enlightenment' or new general understanding as much as it might address a specific problem or issue [16, 64]. The exploitation of existing knowledge through policymaking can thus include expansive and diffuse forms of valorisation, conceptualised through general principles such as societal well-being or public value, rather than being limited to the production of directly quantifiable impacts [65].

In summary, well-developed bodies of literature about knowledge production, translation, communication, and utilization processes deal with common themes of interactions among different types of actors, ways of organising that institutionalise interconnections between scientific and societal actors and entities, and the mobilisation of the outputs of scientific research to generate value. We use these literatures as bases for the development of our conceptual model, which includes a selection of components that forge connections among researchers, research results, stakeholders, and citizens in various ways, contributing to the ceaseless flows of knowledge that shape and re-shape both science and society. In the following section we describe our conceptual approach and specify the components of our model.

## Our model of value generation from research

In this paper we adopt a broadly constructivist perspective that value from research is actively produced in specific contexts. We understand that evolving constellations of scientific and societal actors will conceptualise and realise value in different ways at different times and in different situations [66]. The identities of actors may shift from being knowledge producers to research users to beneficiaries and so on, depending on their relational positioning in these different contexts. The assumption therefore is not that transcendent scientific knowledge is produced in splendid isolation and then deployed in society, but rather that societal actors and influences are always-already entwined in the dynamic processes of scientific research and science-based innovation. Eliminating an arbitrary distinction between science and society avoids what Bozeman and Sarewitz [1] describe as the overemphasis on academic (inside/science) and economic (outside/society) conceptions of research value, to the exclusion of other forms of (public) value. Instead, we are encouraged to focus on the ways in which science and society are themselves configured by their mutual entanglement and ongoing struggles over what types of value can, or should [67], be realised from research. We understand research value to be generated actively through the multiple research-based knowledge processes that produce and entwine science and society.

The multiple research–related knowledge processes relevant to the generation of value from research involve mixed sets of actors with diverse interests and objectives. These processes operate concurrently, both inter- and independently, to configure what value is realised from research. We therefore abandon stylised sequentialist explanations of how value is produced from research in society, instead considering that all relationships and processes that contribute to both scientific and societal value creation involve multi-directional flows of relevant knowledge and information in a multitude of different forms.

Our approach also steps away from utilitarian and pragmatist approaches to the generation of 'societal impact' from research [51], instead seeking to engage with the uncertainty, indeterminacy, and ambiguity of research valorisation processes. First, there is uncertainty regarding the *scope* of research valorisation. Valorisation may occur at different points or moments within research-based knowledge processes: during knowledge creation, through awareness of findings, use of research, or in exploitation of socio-economic benefits [68]. It also encompasses a wide range of research-related activities including publishing findings, transmitting results, collaborating with practitioners, or integrating research into work practices or public policies [69]. Second, the *procedure* through which research value is realised is characterised by indeterminacy. Scientific researchers are often encouraged to directly target their research strategies and actions at the generation of societal gains, framing researchers as being accountable for the knowledge outputs over which they have direct control [70]. However, research results also follow alternative and unforeseeable paths that can contribute to the emergence of societal benefits, independent of researchers' intentional control or influence [71]. Third, the

**Table 1. Summary of the research-related knowledge processes approach to value creation.**

| Scope | Varied types of research-related activities | Co-existing research-based knowledge processes | *Multi-dynamic* |
|---|---|---|---|
| | Diverse actors engaged in research-related activities | Constructivist spectrum of valorisation possibilities | |
| Procedure | Direct intentional action by researchers, societal actors | Interactive processes within relational structures | *Multi-directional* |
| | Indirect contingent and accretive contributions | Ceaseless knowledge flows | |
| Evidence | Complementary datasets | Interchangeable networks of data | *Multi-faceted* |
| | Contextualised indicators | Sensitivity at different scales | |

*evidence* used to apprehend and attribute value from research produces ambiguous results. The further one moves away from the context of a particular research-based intervention the more likely that the attribution of impacts is overdetermined by multiple intruding factors [12]. Rather, multi-faceted networks of data [8] are needed to reflect the complex interrelations and dynamics of multiple research-related knowledge processes operating in specific contexts [70, 72]. In recognition of these challenges, our model includes a range of concurrent and mutually influential components without ranking their importance or ordering them into a sequence.

The empirical research question that guides our model-building work and our empirical exploration is: how are research-related knowledge processes involved and related in generating value for science and society? In this paper, our attempt to shed some light on this question is limited to the co-existence of the different components of research-based knowledge processes included in our model. We investigate the co-presence of these components and describe the archetypal patterns that emerge. Table 1 summarises our approach.

Our conceptual model includes nine value components that represent a 'quadruple helix' of research-related knowledge processes: production, translation, communication and utilization. These components are conceptually and analytically distinguishable aspects of the complex ways in which science and society are intermeshed, as described in Table 2.

**Table 2. Value components: Description and predominant research-related knowledge processes.**

| # | Description of value components | Processes |
|---|---|---|
| C1 | **Commercialisation** processes include practices specifically related to transforming scientific knowledge into marketable products or services, or industrial processes, with the ultimate objective of creating profitable applications. | Translation |
| C2 | **Dissemination** refers to the circulation of research results by a range of research actors, including researchers, research centres/institutes, faculties, universities, and scientific publishers, in the interests of promoting these results as widely as possible to potential research users. | Communication Translation |
| C3 | **Engagement** includes formal and informal productive interactions between researchers and societal agents, including non-academic actors such as firms, government agencies, non-profit organisations and citizens. | Production Translation |
| C4 | **Joint research** refers to fundamental collaborative work involving researchers and non-academic partners to design and perform knowledge production activities. | Production Translation |
| C5 | **Media promotion** includes traditional broadcast or publishing media, such as movies, television, newspapers, or magazines that convey information to the public in different manners. | Communication |
| C6 | **Openness** refers to modes of access and participation for scientific and societal stakeholders in all research-related knowledge processes. | All |
| C7 | **Public policy** refers to the take-up of research results in the fields of public administration, government, health, etc. | Communication Utilization |
| C8 | **Social visibility** refers to the amplification of research and research results through social media, particularly their independent circulation by citizens, citizen organisations or other interest groups. | Communication |
| C9 | **Transmission** refers to researchers' direct promotion of research results in tailored non-academic form to potential end-users. | Translation Communication |

Each of our components is understood to rely on predominant key actors and stakeholders; yet no scientific or societal actors are necessarily excluded from any component. This is a recognition that high degrees of fluidity exist between these components in many cases. For example, *joint research*, *engagement*, and *commercialisation* will often be intertwined in actual work practices. Other components, such as *openness*, do not fit neatly within a single main research-related knowledge process but are transversal.

At the same time, we recognise that there are empirically observable patterns involving sets of actors and actions that operate at different scales (projects, programmes, missions, challenges, etc.) to generate value from research. Scientific and societal actors often organise themselves according to certain scripts or templates that structure the process of conceptualising and realising value from research, such as lengthy and expensive pharmaceutical drug development pipelines, for example. However, there is an increasing recognition that such scripts have too often been viewed as largely rational and technical procedures toward principally technological development, that actively purify social, ethical, and political aspects from struggles to conceptualise and realise research value [2, 66]. Our model neither privileges any of its individual components, nor prioritises technological development over its social acceptance. We do recognise that some of the components can be considered to mobilise and organise 'typical' networks of actors and familiar forms of distributed agency. We would expect to see some evidence of this in our empirical results.

Table 3 expands on our model components in terms of selected characteristics that shape value creation. The predominant research-related knowledge processes in each component include main societal agents involved, the focus of knowledge flows, key mechanisms by which knowledge is mobilised, and typical potential outcomes.

## Methodological design

### Data description

The aim of our experimental empirical analysis is to explore relations between the components of our model. To do this we combine data related to individual researchers and to published

**Table 3. Characteristics of each component in our value model.**

| # | Value component | Predominant research-related: | | | | |
|---|---|---|---|---|---|---|
| | | *Knowledge processes* | *Societal actors* | *Knowledge flows* | *Mechanisms* | *Outcomes* |
| C1 | Commercialisation | Translation | Industry partners | Knowledge application | R&D+I | Industrial processes Marketable products |
| C2 | Dissemination | Communication Translation | Potential research users | Knowledge diffusion | Use of communication tools | Circulation of research results |
| C3 | Engagement | Production Translation | Stakeholders | Knowledge exchange | Productive interactions | Collaborative networks |
| C4 | Joint research | Production Translation | Non-academic R&D partners | Knowledge production | Co-production | Novel results/processes Other outputs |
| C5 | Media promotion | Communication | Media professionals Public audiences | Knowledge conveyance | Broadcast of research / results | Increased public attention Societal awareness |
| C6 | Openness | All | Potential research users | Knowledge accessibility | Digital infrastructure Data sharing, open access | Re-use of academic research/research results |
| C7 | Public policy | Communication Utilization | Public administration Government | Knowledge creep | Scanning, accretion | Influence on policy debates/ decisions |
| C8 | Social visibility | Communication | Interest groups Citizens | Knowledge amplification, prominence | Mentions, likes, shares | Increased attention Sharing of interests |
| C9 | Transmission | Translation Communication | Research users Beneficiaries | Knowledge usability | Tailored research findings | Improved research absorption |

| Steps | Data retrieved | Source |
|---|---|---|
| Identification of researchers affiliated to Spanish institutions based on publication data | 57,406 researchers using the CvE algorithm (Caron & van Eck, 2014) | Web of Science |
| Survey to researchers between June and July, 2016 | 11,992 valid responses received with a 21% response rate | Survey |
| Retrieval of publications for the 2013-2015 period | 83,521 publications from 11,419 researchers | Web of Science |
| Download mentions to publications on Oct, 2017 | 71,163 publications with at least one mention in Twitter, news media and policy docs from 9,190 researchers | Altmetric.com |
| Identification of Open Access status on April, 2019 | 71,163 publications identified, 49% available in Open Access | Unpaywall |

**Fig 1. Overview of the data retrieval steps, number of records and data source used.**

research outputs to form a hybrid 'network of data' [8] composed of nine variables. To construct our dataset, we gather information about both researchers' activities and interactions (researcher data) and about the visibility and social attention afforded to research publications they have authored (research publication data). Our starting point is a set of researchers affiliated to Spanish organisations in the period 2013–2015, according to their publications in the Web of Science. Fig 1 provides an overview of the data collection process, in which four data sources were combined to create our dataset.

**Researcher data.** Researcher data comes from a survey of scientists affiliated to Spanish institutions according to their publication record in the 2012–2014 period derived from the EXTRA project [73], conceded by the Spanish Ministry of Science, and approved by the Spanish Research Council. The survey took place between June and July 2016, receiving a 21% response rate (11,992 valid responses). Respondents work in all fields of science including engineering and physical sciences (STEM), biology and medicine (BIOMED) and Social Sciences and Humanities (SSH). The data were analysed anonymously. Respondents were asked about research-related activities conducted in the 2013–2015 period. The five survey questions we use and the component of our model to which each relates are specified in Table 4.

**Research publication data.** Research publication data was extracted from the publication record of survey respondents for the same period (2013–2015) from Web of Science. We used the CWTS author name disambiguation algorithm [74], which is considered the unsupervised method yielding the best results to date [75]. Only journal articles and reviews were retrieved, obtaining 83,521 publications for 11,419 of the respondents (95%). Data on social media mentions of these publications were subsequently retrieved from Altmetric.com, one of the main data sources for this type of metrics [76, 77]. Altmetric.com relies on the use of output identifiers (i.e., Digital Object Identifier or DOI) to extract social media mentions; 94.3% of the publications in our dataset included a DOI, hence only these could be queried. Both of these databases are biased towards English language publications and hence, limit any study trying to analyse non-English literature [78, 79].

**Table 4. Variables and data sources, model components.**

| # | Value component | Definition of variables | Source |
|---|---|---|---|
| C1 | Commercialisation | Number of types of commercialization activities (patent licensing, spin-offs) in which researchers have participated | EXTRA survey |
| C2 | Dissemination | Frequency of use of analogue and digital communication tools to spread research findings among potential research users | EXTRA survey |
| C3 | Engagement | Number of different types of stakeholders with whom formal interactions took place (SMEs, government agencies, non-profit organisations) | EXTRA survey |
| C4 | Joint research | Number of different types of non-academic R&D partners with whom joint projects were conducted | EXTRA survey |
| C5 | Media promotion | Share of journal publications that have been mentioned at least once in news media outlets | Altmetric.com |
| C6 | Openness | Share of journal publications which are open access | Unpaywall |
| C7 | Public policy | Share of journal publications that have been cited at least once in a policy brief | Altmetric.com |
| C8 | Social visibility | Share of journal publications that have been mentioned at least once in Twitter | Altmetric.com |
| C9 | Transmission | Frequency of promotion of research use (presentations in non-technical language, demonstrations or discussions with final users) in which researchers have participated | EXTRA survey |

Additionally, we retrieved information on the open access (OA) status of publications from Unpaywall, a search engine which identifies OA versions of scientific literature in the world wide web [80–82]. This source also relies on DOIs, and hence publications without a DOI were removed. The research publication data we use and the component of our model to which each relate are specified in Table 4. After removing all cases with missing data, a set of 9,190 survey records was obtained.

## Operationalization of our model

A range of different variables could potentially be used as measures for each our model components. In this paper, we have operationalized the model by defining a single variable for each component (Table 4) to compile the dataset we use. Table 5 provides descriptive statistics for these variables by major scientific fields (Biomedical-BIOMED; Social Sciences and Humanities-SSH; Science, Technology, Engineering and Maths-STEM), based on survey respondents' self-reported field.

**Table 5. Descriptive statistics, model components by scientific field.**

| | | BIOMED | | | SSH | | | STEM | | |
|---|---|---|---|---|---|---|---|---|---|---|
| # | Value component | n | mean | sd | n | mean | sd | n | mean | sd |
| C1 | Commercialisation | 2,130 | 0.52 | 1.02 | 1,950 | 0.20 | 0.57 | 5,110 | 0.60 | 1.02 |
| C2 | Dissemination | 2,130 | 1.53 | 0.66 | 1,950 | 1.76 | 0.76 | 5,110 | 1.52 | 0.65 |
| C3 | Engagement | 2,130 | 1.18 | 1.31 | 1,950 | 1.23 | 1.37 | 5,110 | 1.21 | 1.29 |
| C4 | Joint research | 2,130 | 0.84 | 1.15 | 1,950 | 0.98 | 1.11 | 5,110 | 0.79 | 1.09 |
| C5 | Media promotion | 2,130 | 0.05 | 0.11 | 1,950 | 0.02 | 0.10 | 5,110 | 0.02 | 0.07 |
| C6 | Openness | 2,130 | 0.45 | 0.33 | 1,950 | 0.36 | 0.38 | 5,110 | 0.43 | 0.35 |
| C7 | Public policy | 2,130 | 0.01 | 0.06 | 1,950 | 0.01 | 0.07 | 5,110 | 0.00 | 0.03 |
| C8 | Social visibility | 2,130 | 0.47 | 0.31 | 1,950 | 0.25 | 0.32 | 5,110 | 0.24 | 0.27 |
| C9 | Transmission | 2,130 | 2.64 | 0.97 | 1,950 | 2.87 | 0.97 | 5,110 | 2.77 | 0.94 |

BIOMED: Biomedical; SSH: Social Sciences and Humanities; STEM: Science, Technology, Engineering and Maths.

## Statistical design

The empirical analysis is structured in three parts. First, we present a descriptive analysis in which we explore the relations between the different components of our model. We investigate the distribution of each variable and search for correlations between them. Second, we look for patterns among these variables. We perform an archetypal analysis which aims at identifying prototype configurations according to the intensity of each variable. Third, we relate the resulting archetypes with research topics, by overlaying archetype scores on science maps. Analyses were conducted using the R statistical programming language [83]. More specifically, the ggplot2 and ggally packages [84, 85] were used for visualisations, while the archetypes package was used for conducting the archetypal analysis [86]. The script used is openly accessible at https://github.com/elrobin/value-components and the codebook is available at https://rpubs.com/elrobin/value-components. Overlay co-word maps were created with VOSviewer [87]. The dataset used is openly accessible [88]

The archetypes created are extreme observations in a multivariate dataset, representing convex combinations of the observations that result from a least squares problem [89]. In contrast to clustering techniques, archetypal analysis does not aim at classifying, but provides an overview of the values of prototypes and assigns to each observation an $\alpha$ score for each of the identified archetypes. Therefore, each observation is represented as a mixed composition of these archetypes, some resembling the archetypes more than others. This is not the first study using this technique in the field of scientometrics. It has been previously applied to identify types of researchers based on their publication and citation performance [90], and to identify profiles of researchers based on their contribution statements [91].

Given a multivariate dataset with $n$ observations and $m$ variables, where $n$ denotes the researchers in our dataset and $m$ the nine components of our model, $X$ is a $n \times m$ matrix of archetypes. Then, the residual sum of squares (RSS) is denoted by

$$RSS = ||X - \alpha Z^T||_2,$$

with $Z = X^T \beta$, where $\alpha, \beta$ are positive coefficients and $||\cdot||_2$ denotes the Euclidean matrix norm. Each observation is therefore represented as a convex combination of archetypes

$$X \approx \alpha Z^T.$$

One of the advantages of this approach is that archetypes are neither forced to be mutually exclusive nor to remain the same when changing the number of archetypes considered. That is, each observation is assigned with an $\alpha$ score for each of the archetypes produced. $\alpha$ scores show the closeness each observation has to a given archetype and range between 0 and 1, being 1 a complete resemblance with a given archetype and 0 no resemblance. Hence, while in some cases an observation may clearly be identified with one specific archetype, in other cases, the observations may reflect a mix or configuration of different archetypes. The appropriate number of archetypes is identified by following an elbow criterion based on the RSS obtained for each number of archetypes.

Finally, we look specifically into the topic contents of the research publications in our dataset and their relationship with the archetypes identified. This is done by mapping the terms used in the titles of the publications based on their co-occurrence. Co-word maps are a well-established visualisation approach used in the field of scientometrics [92]. Overlay science mapping was introduced as a means to establish comparisons between different local maps which are overlaid on a global map or base map [93]. We overlay the $\alpha$ scores obtained by each observation on the terms extracted from their publication titles. In our application of the technique, the 'global map' is represented by the scientific field being portrayed, while 'local map'

refers to each of the archetypes identified in that field. We include only terms which occur at least 10 times. For visualisation purposes only the 60% most relevant terms are displayed.

## Findings of the empirical analysis

### Components of the model

Fig 2 shows how the component variables are distributed and how they relate to each other. As can be observed, distributions are relatively similar across fields. These are highly skewed with extreme outliers and many zero values in the cases of *media promotion* and *public policy*. Notable exceptions can be found in the cases of *openness*, with a relatively homogeneous distribution, and *transmission*, which seems to follow a normal distribution.

While most components showed either low or no correlation with other components, there are some notable exceptions. *Joint research* and *engagement* are strongly and positively correlated with each other, with all values above 0.7. This is understandable as ongoing informal processes of engagement are often a pre-condition for the formalisation of joint research projects, for example. *Commercialisation* and *transmission* show a low but positive correlation with *engagement* (between 0.33 and 0.33). This is also evident between *transmission* and *dissemination* (0.32).

### Archetypes and epistemic relations

This section presents the archetypes emerging from our analyses and the relation each archetype has with the topics of research publications. Fig 3 shows the three archetypes identified for the entire set of researcher/research publication cases, along with the overlay science map for each archetype. The parameters of each archetype are shown on the left side of the (Fig 3A, 3C and 3E), with the corresponding science overlay map for each archetype on the right side (3B, 3D, and 3F). As can be observed, topic terms are clustered into three groups within each overlay map. The group at the centre-top of the map includes terms related to STEM fields, such as 'catalyst', 'ligand', and 'nanocomposite'. The group to the right side of the map is prominently related to high energy physics as it includes terms such as 'atlas detector',

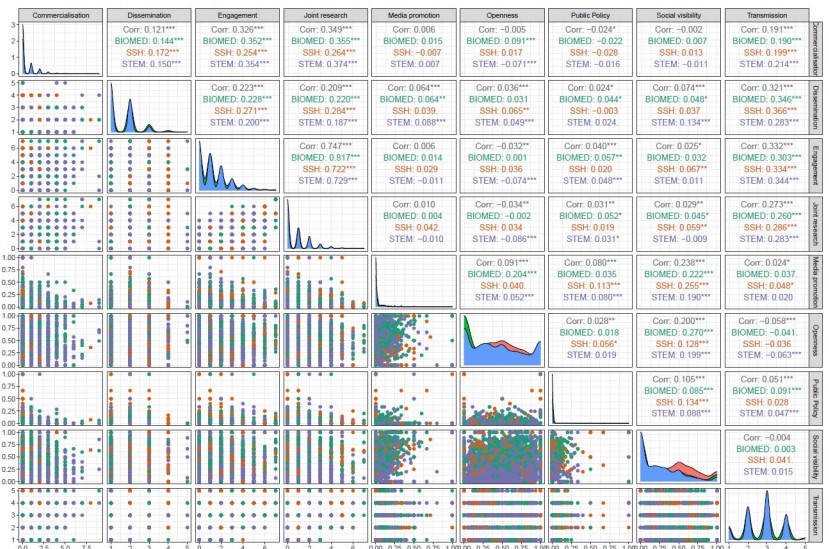

**Fig 2. Distribution patterns and correlation matrix for the nine components overall and by field.** BIOMED: Biomedical; SSH: Social Sciences and Humanities; STEM: Science, Technology, Engineering and Maths.

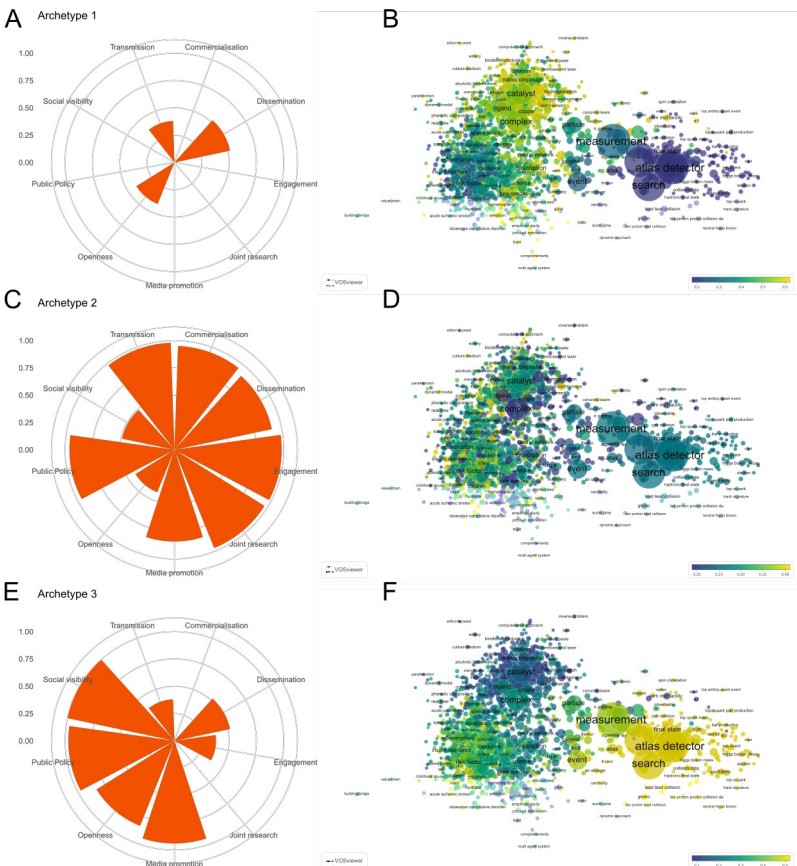

**Fig 3. Parameters and topic maps for three archetypes in all fields of science.** Archetypes that emerge for all cases (A, C and E), the variables transmission, commercialisation, dissemination, engagement and joint research have been rescaled (0–1) to allow comparisons with the rest of the variables. Archetype parameters are shown in percentiles. Overlay maps of α scores corresponding to words included in the titles of research publications (B, D and F). Words follow a colour grading based on their maximum or minimum value, the extreme values being yellow (for those present in publication titles of cases resembling a given archetype) and blue (those which are not present in the publication titles of cases resembling the archetype). An online version of this map is available at: http://sl.ugr.es/allarchetypes.

'measurement', and 'final state'. Finally, the group on the left side includes terms mainly related to biomedical fields including 'inflammation', 'risk factor', and 'insulin resistance'. Terms from the SSH are quite hidden, probably due to the relatively low number of publications included compared to STEM and biomedical fields.

The three model components common to all three archetypes generated are *dissemination*, *openness*, and *transmission*. Archetype 1 (Fig 3A) is characterised by relatively weak values for these three components and no other components are present. Based on the topics of research publications as visualised at the yellow end of our colour spectrum (Fig 3B), this archetype seems to be composed mainly by the STEM fields in the top-centre of the map, although some medical and human sciences are also apparent in the lower part of the topic map.

Archetype 2 shows high values from multiple components including *commercialisation*, *dissemination*, *engagement*, *joint research*, and *transmission* (Fig 3C). *Media promotion* and *public policy* are also prominent components of this archetype. The map for this archetype is the least strongly defined by specific topic areas. The terms most strongly corresponding to Archetype 2 belong to biomedical fields in the left of the map (Fig 3D), with some topics in other STEM

fields also apparent in the upper part of the map. Archetype 3 (Fig 3E) is characterised by high *media promotion*, *public policy, and social visibility*, and relatively high *public policy* components. Lower values, on a par with those for Archetype 1, are also present for the *dissemination* and *transmission* components. This archetype appears the most field specific in terms of the topic map (Fig 3F), with topics related to high energy physics prominent to the right of the map. Some biomedical topics are also apparent in the lower left of the map.

Figs 4 to 6 reproduce the same visualisations as in Fig 3, but now focusing on archetypes within each of the three major fields we analyse. Fig 4 focuses on STEM fields, where we also observe three archetypes. Two distinct groups can be observed in the research topics map, with a cluster of high energy physics topics to the right of the map relatively separated from other STEM fields. Archetype STEM1 (Fig 4A) corresponds to this high energy physics cluster of topics (Fig 4B). This archetype has high values for *dissemination*, *media promotion*, *public policy*, and *social visibility*, and medium values for *commercialisation*, *openness*, and *transmission*.

Archetype STEM2 (Fig 4C) shows high values for *commercialisation*, *engagement*, *joint research*, and *transmission*. Medium to high values are also evident for the *dissemination* and

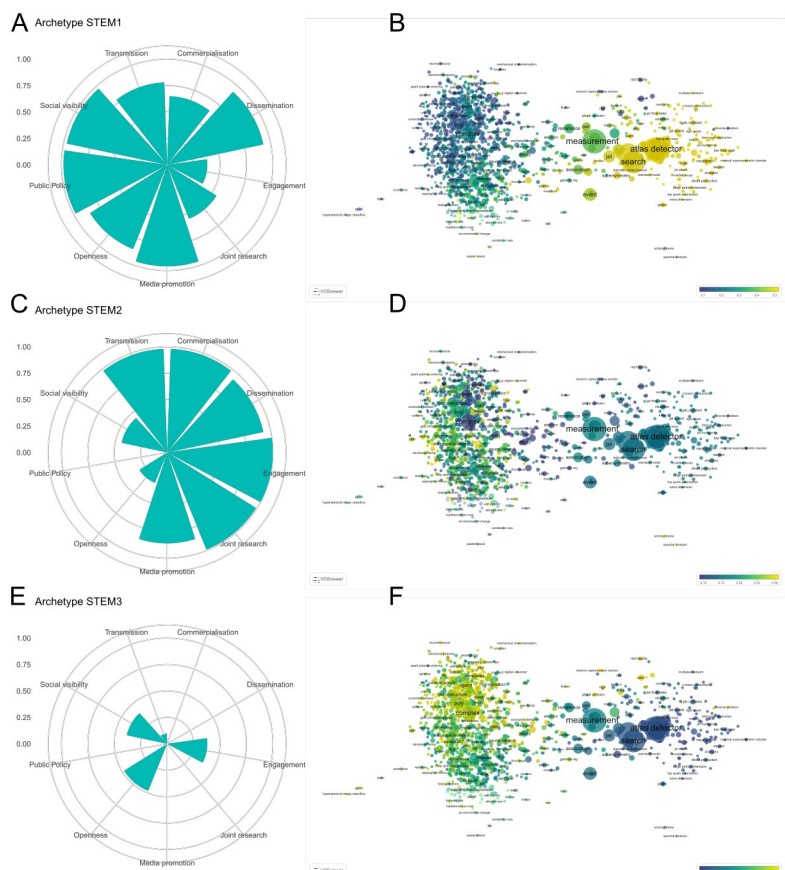

**Fig 4. Parameters and topic maps for three archetypes in STEM field.** Archetypes that emerge for all STEM cases (A, C and E), along with overlay maps of α scores corresponding to words included in the titles of research publications (B, D and F). Archetype parameters are shown in percentiles. Words follow a colour grading based on their maximum or minimum value, the extreme values being yellow (for those present in publication titles of cases resembling a given archetype) and blue (those which are not present in the publication titles of cases resembling the archetype). An online version of this map is available at: http://sl.ugr.es/stemarchetypes.

*media promotion* components. The overlay topic map for this archetype is less clearly defined and corresponds to across a wide range of research fields, with prominent topics including 'mass spectrometry' and 'gas chromatography' (Fig 4D). Archetype STEM3 is characterised by a relatively low values for *engagement*, *openness*, and *social visibility*. This archetype corresponds to topics including 'microstructures', 'ligands', and 'hydrogenisation', as observed in Fig 4F.

Fig 5 showcases the SSH field, in which three archetypes are also identified. The overlay map for SSH forms a single cluster in which human health and development appear prominent on the left side, with socio-economic fields prominent on the right. Archetype SSH1 is characterised by high levels of *media promotion*, *public policy*, and *social visibility*, along with a medium level of *openness* (Fig 5A). The overlay science map (Fig 5B) for this archetype (which is based on a relatively low number of research publications) corresponds to health related topics including 'disability' and 'polymorphism'. Archetype SSH2 includes medium-low values for *dissemination*, *engagement*, *openness*, *social visibility*, and *transmission* (Fig 5C). The associated topic map (Fig 5D) highlights topics including 'innovation', 'industry', 'technology', and 'productivity'. Archetype SSH3 (Fig 5E) is composed of high values for *commercialisation*,

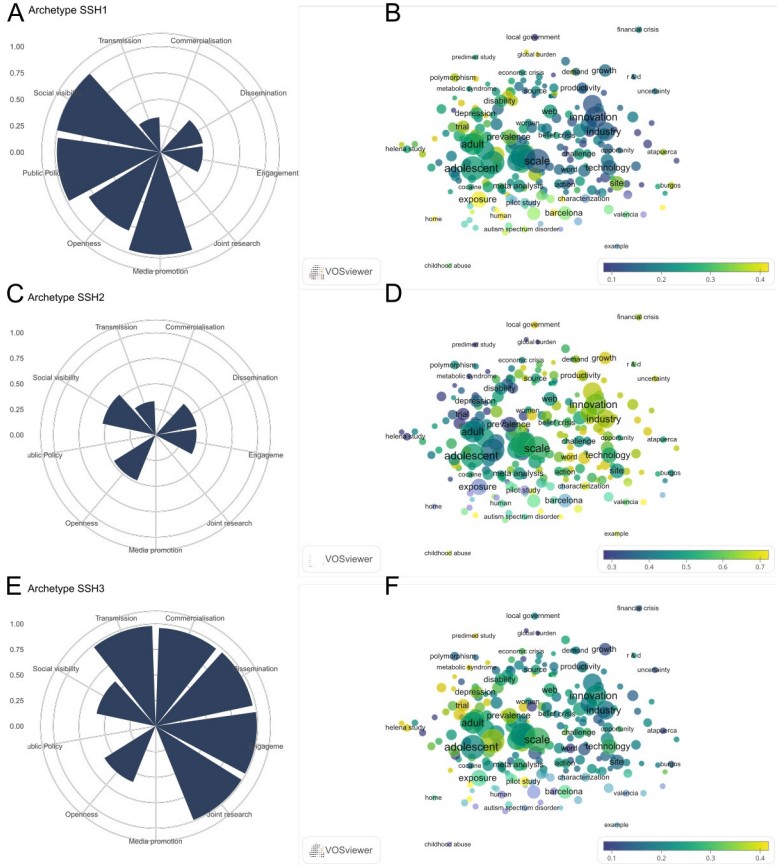

**Fig 5. Parameters and topic maps for two archetypes in SSH field.** Archetypes that emerge for all SSH cases (A and C), along with overlay maps of α scores corresponding to words included in the titles of research publications (B and D). Words follow a colour grading based on their maximum or minimum value, the extreme values being yellow (for those present in publication titles of cases resembling a given archetype) and blue (those which are not present in the publication titles of cases resembling the archetype). An online version of this map is available at: http://sl.ugr.es/ssharchetypes.

*dissemination*, *engagement*, *joint research*, and *transmission* components, along with medium values for *openness* and *social visibility*. This archetype is associated with topics related to adolescence, health and development (Fig 5F).

Three archetypes are also observed in the field of BIOMED (Fig 6). The corresponding overlay maps are clustered around species and diversity topics to the right, hospitals and reviews of medical knowledge in the centre, and surgery and specific human health conditions to the left. Archetype BIOMED1 (Fig 6A), includes high values for *commercialisation*, *engagement*, *joint research*, *and transmission* components, along with high-medium values for *dissemination* and *public policy*. This archetype corresponds to topics in the centre of the science map (Fig 6B). Archetype BIOMED2 (Fig 6C) is composed of low-medium levels of the *engagement* and *transmission* components and low values for *openness* and *social visibility*. The topic overlay map (Fig 6D) for this archetype highlights 'species' and diversity topics to the right of the map, but 'surgery' and 'cirrhosis' topics on the lower left are also prominent. Archetype BIOMED3 (Fig 6E) includes high values for *openness* and *social visibility* components, high-medium values for *media promotion* and *public policy*, and medium values for *commercialisation* and *dissemination*. The research topic map (Fig 6F) produced for this archetype is not as

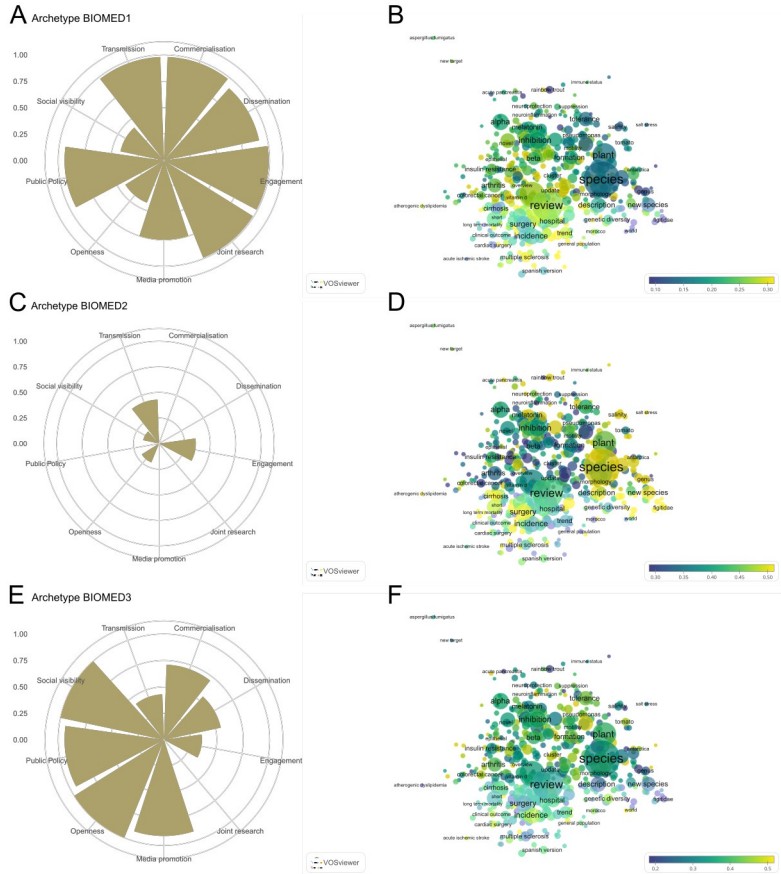

**Fig 6. Parameters and topic maps for three archetypes in BIOMED field.** Archetypes that emerge for all biomedical cases (A, C and E), along with overlay maps of α scores corresponding to words included in the titles of research publications (B, D and F). Words follow a colour grading based on their maximum or minimum value, the extreme values being yellow (for those present in publication titles of cases resembling a given archetype) and blue (those which are not present in the publication titles of cases resembling the archetype). An online version of this map is available at: http://sl.ugr.es/biomedarchetypes.

clearly clustered, with 'pseudomonas' among associated topics mainly located in the upper half of the map.

## Discussion

In this paper we presented a model of value generation from research-related knowledge processes that entwine and co-produce science and society. The components we used in our model were aspects of processes of knowledge production, translation, communication, and utilization that have been identified as important by previous research. The empirical experiment the paper reports sought to explore the use of archetypal analysis for operationalizing our conceptual model. We constructed this experiment at an intermediate level of analysis, using information about the co-production of knowledge, about the translation and application of research findings, and information about how research results circulate and are used. The archetypal relations we uncovered illustrate the different configurations of research-related knowledge processes that stretch across science and society. We also emphasized that this research is exploratory, due to the acknowledged limitations of some of the data proxies we used to measure our model components.

The analysis addressing our empirical research question produced archetypal configurations made up of distinctive combinations of our model components. These archetypes map onto different patterns of topics on our science maps. The three archetypes that were generated for our entire dataset (Fig 3) were highly distinctive. These three archetypes remained quite stable in our further analyses at the level of the STEM, SSH, and BIOMED fields.

A first noticeable feature of our results is an archetype composed of five of our model components: *commercialisation*, *dissemination*, *engagement*, *joint research*, and *transmission*. These components feature together, with varying individual values, in all our archetypal comparisons (Archetype 2 all fields, STEM 2, SSH 3, BIOMED 1). Strong values for *commercialisation* activities are accompanied by equally strong values for *joint research* performance and *transmission* activities, in which research results are presented to potentially interested stakeholders in a specifically tailored format. Strong values are also present in this archetype for *engagement* practices and *dissemination* activities. The components of this archetype appear coherent with the literature on the outcomes of engaged co-production activities between researchers and non-acadamic actors, their translation toward markets and communication to end-users (Table 3). This result can thus be considered an initial indicator of confidence in our methodological approach.

A second noticeable feature is the emergence of an archetypal configuration with strong values for *media promotion*, *openness*, *public policy*, and *social visibility*. These components feature together, with varying individual values, in all our archetypal comparisons (Archetype 3 all fields, STEM 1, SSH 1, BIOMED 3). The *dissemination* component is associated with this archetype at field level, showing strong values in STEM 1 and medium values in SSH1 and BIOMED 3. This archetype has a consistency around components related to the accessibility, communication and take up of knowledge.

While these two noticeable archetypal configurations are distinctive, *media promotion* and *public policy* components are features of both at the level of all fields (Fig 3) and in BIOMED1 (Fig 6A). Other features of the results include the presence in all of our archetypal comparisons of a third residual configuration. This archetype mainly features *dissemination*, *openness*, *social visibility*, and *transmission* components, but with relatively weak values. The one consistent component across these configurations is openness.

From a methodological perspective to our knowledge no other study in the field has analysed a hybrid dataset resulting from the integration of bibliometric, OA, altmetric, and survey

data. The use of archetypal analysis to treat this dataset had the advantage of enabling us to consider the nine components of our value model collectively and not on a one-to-one basis. In fact, the correlation matrix shows that if such a common approach had been followed, we would have not been able to provide much evidence on the relation between components (Fig 2).

There are some limitations to exploratory research of this type. First, there are important processes associated with knowledge utilisation–such as end-user innovation–that are not part of our model. The model is currently limited by the fact that research value emerging in use, and its interplay with local and indigenous knowledges for example, is not adequately represented. This is a task for future model-building work. Second, our empirical experiment uses data based on researchers and research publications affiliated to research performing organisations in Spain. The influence of national system characteristics will affect our substantive results, at this stage to an unknown extent. Nevertheless, much scientific knowledge production is the result of international collaboration which likely moderates this influence, not least by expanding the space of knowledge-related research processes to collaborator countries.

Third, the variables and measures used in our empirical experimentation are not necessarily the best that can be found or constructed to operationalize our model components. The measures used derive from our own empirical research and from large-scale publicly available sources and are largely accessibility-related choices. Due to these limitations we have been careful to describe this research as exploratory, yet we are satisfied the substantive results obtained are coherent in terms of our model characteristics (Table 3) and constitute a sound proof of concept. Refining the model and identifying better quality data points that can be accessed, or could be created, is part of future work. For example, the recent launch of Overton, a database specifically focused on the identification of mentions of scientific literature in policy documents, could provide better coverage, yielding more meaningful results in quantitative analyses such as ours [94]. Developing components and data that can allow us to better incorporate knowledge utilization processes in our approach is an important future challenge. Fourth, the empirical experiments conducted in this paper may not be of the optimum scale to best profit from the archetypes methodological approach. We conducted our experiments at a scale that enabled us to illustrate the potential of the approach to identify patterns within the profusion of knowledge processes that produce value in science and society. However, other levels of analysis may be more effective, including those relying (partly) on automated machine learning approaches.

From a policy perspective, our results are interesting in two main ways. First, constructing a network of data allowed us to explore simultaneously a number of different processes relevant to how research value is generated. We were able to identify different configurations of our model components that map onto different parts of the topic spectrum in science. This suggests that certain processes and their combinations may be of particular importance in generating value from different areas of research. Second, our work provides a different angle on the interactions between science and society that can be understood as potential drivers for generating value from research. Whereas most academic approaches seek to establish causal sequences or effect chains running through knowledge production and translation to societal impact, we modelled our components as concurrent and co-evolutionary elements of a multi-dimensional interactive system. In doing so we found strong similarities and only minor differences in the structure of the archetypal configurations for different major fields of science. Some archetypes were suggestive of institutionalised scripts that are well-studied in the existing literature, whilst others are less easily interpreted and might suggest a gap in our understanding. A closer examination of the research topics in different major fields that correspond to similar archetypal forms might offer an interesting avenue for further exploration in this regard.

## Author Contributions

**Conceptualization:** Irene Ramos-Vielba, Nicolas Robinson-Garcia, Richard Woolley.

**Data curation:** Nicolas Robinson-Garcia.

**Formal analysis:** Irene Ramos-Vielba, Nicolas Robinson-Garcia, Richard Woolley.

**Funding acquisition:** Nicolas Robinson-Garcia.

**Investigation:** Irene Ramos-Vielba, Richard Woolley.

**Methodology:** Irene Ramos-Vielba, Nicolas Robinson-Garcia, Richard Woolley.

**Software:** Nicolas Robinson-Garcia.

**Supervision:** Richard Woolley.

**Visualization:** Irene Ramos-Vielba, Nicolas Robinson-Garcia.

**Writing – original draft:** Irene Ramos-Vielba, Richard Woolley.

**Writing – review & editing:** Nicolas Robinson-Garcia, Richard Woolley.

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
