## [Decision Letter · Decision Letter 0]

5 Oct 2021

PONE-D-21-16293A value creation model from science-society interconnections: Components and archetypesPLOS ONE

Dear Dr. Robinson-Garcia,

Thank you for submitting your manuscript to PLOS ONE. After careful consideration, we feel that it has merit but does not fully meet PLOS ONE’s publication criteria as it currently stands. Therefore, we invite you to submit a revised version of the manuscript that addresses the points raised during the review process. Please submit your revised manuscript by Nov 19 2021 11:59PM. If you will need more time than this to complete your revisions, please reply to this message or contact the journal office at plosone@plos.org. Please include the following items when submitting your revised manuscript:A rebuttal letter that responds to each point raised by the academic editor and reviewer(s). You should upload this letter as a separate file labeled 'Response to Reviewers'.A marked-up copy of your manuscript that highlights changes made to the original version. You should upload this as a separate file labeled 'Revised Manuscript with Track Changes'.An unmarked version of your revised paper without tracked changes. You should upload this as a separate file labeled 'Manuscript'.

We look forward to receiving your revised manuscript.

Kind regards,

Lutz Bornmann

Academic Editor

PLOS ONE

“This research was supported by the Spanish Ministry of Economy, Industry and Competitiveness through the State Plan of Scientific and Technical Research and Innovation (EXTRA project, grant CSO2013-48053-R) and by the Oslo Institute for Research on the Impact of Science (OSIRIS, grant 256240) funded by the Research Council of Norway. Nicolas Robinson-Garcia is currently supported by a Ramón y Cajal grant from the Spanish Ministry of Science (RYC2019-027886-I).”

Reviewers' comments:

Reviewer's Responses to Questions

**Comments to the Author**

1. Is the manuscript technically sound, and do the data support the conclusions?

Reviewer #1: Yes

Reviewer #2: Partly

2. Has the statistical analysis been performed appropriately and rigorously? 

Reviewer #1: I Don't Know

Reviewer #2: Yes

3. Have the authors made all data underlying the findings in their manuscript fully available?

Reviewer #1: Yes

Reviewer #2: Yes

4. Is the manuscript presented in an intelligible fashion and written in standard English?

Reviewer #1: Yes

Reviewer #2: Yes

5. Review Comments to the Author

Reviewer #1: The manuscript "A value creation model from science-society interconnections: Components and archetypes" presents a mixed-data analysis with data from a questionnaire, Altmetric.com, and Unpaywall. The different types of data are assumed to be proxies of nine different value components. Overall, the manuscript is well written and should be of interest to the readers of PLoS One. However, some issues should be resolved before publication. I am unsure how good the different value components are represented by the proxy data. Actually, I am quite skeptical about the assumption that they are good proxies. The authors seem to share this skepticism as can be seen from parts of the Discussion section. Therefore, this study seems to be very explorative in its nature. This should be stated prominently in the manuscript. More detailed comments follow below.

In lines 234-236, the authors state: "As represented in Figure 1, we understand research value to be generated actively through the multiple research-based knowledge processes that produce and entwine science and society." I do not see that Figure 1 is a good representation for this statement. I do not find Figure 1 helpful in its current form.

Regarding the data (lines 330-343) I wonder if "records" (n=6,174) in line 342 refers to "publications" or some other kind of records. If "records" refers to "publications", a very low percentage of the total amount of publications retrieved (n=83,551) could be analyzed. Even if "records" refers to "respondents" (n=11,419), just more than half of the respondents data were analyzed.

In line 363, the authors state that they used R for analysis. Usage of R should be cited. For increasing reproducibility and benefit for the readers, I think that the authors should share their R scripts as supplemental information.

In lines 383-384, the authors state: "One of the advantages of this approach is that archetypes are neither forced to be mutually exclusive nor to remain the same when changing the number of archetypes considered." I do not see why this is advantageous.

The description of the identification of archetypes is not clear enough.

Figures 4-7 contain VOSviewer overlay maps on the right panels. The color codes are explained only in part. It is explained what blue and yellow colors mean but not the colors in-between. However, I think that the explanation of the colors is inaccurate. It puzzles me that the scales have different ranges of values. I suppose that the value of zero corresponds to the absence of a term. What does a value of one mean? Are these terms present in all publication titles? Does a value of 0.5 mean that the term is present in 50% of the publication titles?

It would be helpful for a detailed inspection of Figures 4-7 if web-startable links or the map files of the VOSviewer overlay maps were provided as supplemental information.

Reference 51 contains an invalid DOI.

Reviewer #2: This paper investigates the possibility of detecting typical configurations of science-society interactions in large datasets classified by broad areas of research (STEM, BIOMED, SSH). It is a novel approach. The paper represents an experiment, according to the authors. It deserves to be published, perhaps with a change in title (see below) and an extension at the discussion at the end, the last part starting with “There are some limitations…”. This is the most interesting part. Here, there could be a constructive discussion of why the experiment (in my view) failed.

In short, it fails because the data selected for the analysis cannot meet the demands of the theoretical perspective and the modelling of science-society interactions presented initially. The experiment gets lost within the limitations of the available international data sources in bibliometrics and altmetrics. This new negative result deserves to be published along with concrete ideas of more proper data sources.

The study takes four steps (A-D) that leads to the failure:

The first 8 pages (A) reviews the literature on science-society interactions and establishes a solid and promising theoretical framework that convincingly describes research-related knowledge processes as shaped by interaction.

Then a model for value generation for research (B) is introduced, based on the sequence: knowledge production, translation, communication and utilization. Here, ‘outputs of scientific research’ ‘generate value’. This is a more traditional linear understanding of the relation: knowledge is created within science and then brought out to society. The insights from A in Table 1 seem difficult to represent in Table 2, which better represents B and is more suitable to match with the survey results (C) from the EXTRA project with almost 12,000 responses from Spanish researchers describing their research-related activities.

The introduction of C assumes that summing up numbers of categorized individual activities of Spanish researchers can provide information about how science-society relations are organized in Spain. Can they tell us how clinical research interacts with clinical practice in the health services? Can they tell us how educational research and its organizations in Spain interact with the Spanish school system and its organizations? Can they tell us how geophysics, in an organized way, comes to the aid in the crisis of a volcano outbreak? I think the survey results can only tell us about numbers of pre-categorized activities. Can configurations of field-related typical interactions emerge from this?

As step D, traces of societal interaction are sought in the impact of WoS-publications in altmetric data sources. The study forgets that most publications representing the societal interactions of the same researchers will not be in WoS and they will be in the Spanish language if in publicly available writing at all. The use of WoS-publications and mentions of them in social media is a very limited representation of societal interaction. Societal use of research most of the time does not happen in social media. A sign of the failure in step D is that policy impact is almost invisible in the results. The trouble is that the study is bound up with documents with DOI – we are inside the English-speaking international academic publishing and library world (with only around ten major global publishers defining the DOI) and not out there in the Spanish society.

In the conclusions, I suggest returning to the perspective of A which will lead the way to a much better understanding of science-society relations given that better data sources are found. There is need of a discussion of the limitations of the most used data sources (as in this failed experiment) and proposals for new relevant data sources.

An example: Legal research is mostly nationally oriented and far from good representation in international data sources. However, most countries now have – often on a commercial basis – a continuously updated legal information system that connects between law formation, decisions by legal institutions, legal practice, and legal research. An archetype representing how the interactive science-society relations are typically organized in the legal system of a country might emerge from the use of this data source.

Until the experiment gets closer to a clearer understanding of field-related interactions, I would not talk about archetypes. Archetype is the name of the math and methods in this experiment, not of what you find. I would not use the word in the title. Configurations is a preferable word already used in the text.

6. PLOS authors have the option to publish the peer review history of their article (what does this mean?). If published, this will include your full peer review and any attached files.

Reviewer #1: No

Reviewer #2: No

---

## [Author Response · Author response to Decision Letter 0]

9 Nov 2021

Dear dr. Lutz Bornmann,

Thank you for the opportunity to respond to the two reviews of our manuscript PONE-D-21-16293: ‘A value creation model from science-society interconnections: Components and archetypes’. We are grateful to both reviewers for their constructive comments and suggestions. We also appreciate the reviewers’ willingness to engage fully with the paper, which is a slightly unorthodox mix of conceptual development, methodological experiment, and empirical trial, and for their overall support for our work.

In the following we address, first, the major general concern that both reviewers had with the manuscript. Second, we respond to the specific requests for clarification or change from each reviewer.

The major concern of both reviews is the quality of those data used as proxies for the components in our conceptual model. The reviewers also note that we are not naïve to this issue in the presentation of our work. As they also acknowledge, we make clear cautionary statements about both the quality of our proxies and our desire to improve them in the future, and regarding the potentially considerable margin for improvement in the approximations of our archetypes that better data may produce. The reviewers suggest remedies for this concern related to how the paper is framed. Reviewer #1 would like the paper to be more prominently framed as exploratory in its objectives. Reviewer #2 would like the title changed to reflect archetypes as being our method not our output, more discussion of the limitations of the most used data sources, and if possible, suggestions about new relevant data sources.

In response to these requests, we have added two statements regarding our work as exploratory research, one in the introduction and one in the discussion section. We have changed the title, to refer to archetypes as our method, not as a consolidated empirical output. The discussion of the limitations of the paper, particularly the data proxies, has been strengthened, although we do not wish to speculate about potential future data sources, as to the best of our knowledge such alternatives are currently scarce, with the obvious major exception of the Overton database we included in the manuscript.

In summary, we agree with the main concern of the reviewers, but would also contend that this issue would have been far more serious if the empirical results we obtained did not appear intuitively reasonable and did not allow for coherent interpretation according to our model. Neither reviewer raised any concerns about the ‘configurations’ that emerged from our use of the archetype method. Indeed, we would not have submitted this exploratory research if we did not believe that our results were coherent with our expectations and our conceptual model. Accordingly, we consider our empirical results a proof of concept or exploratory step in the right direction, which will inevitably be improved upon through access to better data proxies and other potential design modifications that we also describe in the final section of the manuscript – such as the potential amenability of the model and method to ‘big data’ approaches. In any case, we would like to point out that putting together a dataset with information from different sources (self-reported, publication and secondary databases) is rare and very few studies can be found in this regard.

The specific comments, requests for clarification of changes from Reviewer #1 included:

• In lines 234-236, the authors state: "As represented in Figure 1, we understand research value to be generated actively through the multiple research-based knowledge processes that produce and entwine science and society." I do not see that Figure 1 is a good representation for this statement. I do not find Figure 1 helpful in its current form.

Figure 1 has been deleted and other figures renumbered.

• Regarding the data (lines 330-343) I wonder if "records" (n=6,174) in line 342 refers to "publications" or some other kind of records. If "records" refers to "publications", a very low percentage of the total amount of publications retrieved (n=83,551) could be analyzed. Even if "records" refers to "respondents" (n=11,419), just more than half of the respondents data were analyzed

We now refer to n=6,174 records as survey records, as they are instances of survey responses and their aggregated publication, OA and altmetric indicators.

• In line 363, the authors state that they used R for analysis. Usage of R should be cited. For increasing reproducibility and benefit for the readers, I think that the authors should share their R scripts as supplemental information

The R scripts have been supplied and R has been cited.

• The description of the identification of archetypes is not clear enough

We have expanded the section on statistical design. Specifically, we have described how archetypes are assigned to observations and how these are interpreted.

• Figures 4-7 contain VOSviewer overlay maps on the right panels. The color codes are explained only in part. It is explained what blue and yellow colors mean but not the colors in-between. However, I think that the explanation of the colors is inaccurate. 

• It puzzles me that the scales have different ranges of values. I suppose that the value of zero corresponds to the absence of a term. What does a value of one mean? Are these terms present in all publication titles? Does a value of 0.5 mean that the term is present in 50% of the publication titles?

We have now included a brief explanation of the grading and maximum and minimum values in order to ease the interpretation of the overlay maps.

• It would be helpful for a detailed inspection of Figures 4-7 if web-startable links or the map files of the VOSviewer overlay maps were provided as supplemental information.

Thank you for the suggestion. These online versions are now available and have been mentioned in each figure’s caption.

• Reference 51 contains an invalid DOI

Many thanks. This has been corrected and the rest of the references have also been revised accordingly.

The specific comments, requests for clarification of changes from Reviewer #2 included:

• The introduction of C assumes that summing up numbers of categorized individual activities of Spanish researchers can provide information about how science-society relations are organized in Spain. Can they tell us how clinical research interacts with clinical practice in the health services? Can they tell us how educational research and its organizations in Spain interact with the Spanish school system and its organizations? Can they tell us how geophysics, in an organized way, comes to the aid in the crisis of a volcano outbreak? I think the survey results can only tell us about numbers of pre-categorized activities. Can configurations of field-related typical interactions emerge from this?

Our data are presented at the broad field level and show the configurations of relations between the different components in our model. These data may be useful for some of the other research questions listed here, although they are not intended for generalization of the Spanish system. Survey results do record numbers of ‘pre-categorized activities’ (i.e., our variable dimensions). There is a rich literature examining configuring configurations of these interactions (which we draw on in part of our model development) that does present results of typical common channels of interactions. The model and analysis presented in our manuscript is complementary to some of these types of studies but is also very different from them.

• The study forgets that most publications representing the societal interactions of the same researchers will not be in WoS and they will be in the Spanish language if in publicly available writing at all. The use of WoS-publications and mentions of them in social media is a very limited representation of societal interaction. … The trouble is that the study is bound up with documents with DOI – we are inside the English-speaking international academic publishing and library world

It is true that the vast majority of the publications we used are written in English. They are also the most important article publications of many of these researchers as, for reasons beyond our control, English remains the hegemonic publication language in most globalized fields of science (certainly in STEM and BIOMED). We agree with the reviewer that there are limitations in terms of language and coverage associated with using the WoS (http://doi.org/10.1023/A:1010549719484) and that improvement in publication databases would enhance the quality of research outputs using these sources. The data proxies for our components, however, cannot inform about representativeness in a given context but they reflect value creation processes through different forms of science-society interconnections.

• [P]olicy impact is almost invisible in the results

The data proxy used for the public policy value component is limited, as we state clearly in the manuscript. We have also strengthened our acknowledgement of the limitations of the data proxies used overall. The emergence of the new Overton database on policy documents, which we also mentioned, is a promising opportunity that we intend to exploit in the future to improve the recognition of policy take-up in our analyses.

We appreciate reviewers’ comments and suggestions, which we have incorporated in the new version of the manuscript.

Sincerely,

The authors

---

## [Decision Letter · Decision Letter 1]

14 Dec 2021

PONE-D-21-16293R1A value creation model from science-society interconnections: Archetypal analysis combining publications, survey and altmetric dataPLOS ONE

Dear Dr. Robinson-Garcia,

Thank you for submitting your manuscript to PLOS ONE. After careful consideration, we feel that it has merit but does not fully meet PLOS ONE’s publication criteria as it currently stands. Therefore, we invite you to submit a revised version of the manuscript that addresses the points raised during the review process. Please submit your revised manuscript by Jan 28 2022 11:59PM. If you will need more time than this to complete your revisions, please reply to this message or contact the journal office at plosone@plos.org. Please include the following items when submitting your revised manuscript:A rebuttal letter that responds to each point raised by the academic editor and reviewer(s). You should upload this letter as a separate file labeled 'Response to Reviewers'.A marked-up copy of your manuscript that highlights changes made to the original version. You should upload this as a separate file labeled 'Revised Manuscript with Track Changes'.An unmarked version of your revised paper without tracked changes. You should upload this as a separate file labeled 'Manuscript'.If applicable, we recommend that you deposit your laboratory protocols in protocols.io to enhance the reproducibility of your results. Protocols.io assigns your protocol its own identifier (DOI) so that it can be cited independently in the future. For instructions see: https://journals.plos.org/plosone/s/submission-guidelines#loc-laboratory-protocols. Additionally, PLOS ONE offers an option for publishing peer-reviewed Lab Protocol articles, which describe protocols hosted on protocols.io. Read more information on sharing protocols at https://plos.org/protocols?utm_medium=editorial-email&utm_source=authorletters&utm_campaign=protocols.

We look forward to receiving your revised manuscript.

Kind regards,

Lutz Bornmann

Academic Editor

PLOS ONE

Reviewers' comments:

Reviewer's Responses to Questions

**Comments to the Author**

1. If the authors have adequately addressed your comments raised in a previous round of review and you feel that this manuscript is now acceptable for publication, you may indicate that here to bypass the “Comments to the Author” section, enter your conflict of interest statement in the “Confidential to Editor” section, and submit your "Accept" recommendation.

Reviewer #1: (No Response)

Reviewer #2: All comments have been addressed

2. Is the manuscript technically sound, and do the data support the conclusions?

Reviewer #1: Yes

Reviewer #2: Yes

3. Has the statistical analysis been performed appropriately and rigorously? 

Reviewer #1: Yes

Reviewer #2: Yes

4. Have the authors made all data underlying the findings in their manuscript fully available?

Reviewer #1: Yes

Reviewer #2: No

5. Is the manuscript presented in an intelligible fashion and written in standard English?

Reviewer #1: Yes

Reviewer #2: Yes

6. Review Comments to the Author

Reviewer #1: I appreciate that the authors shared their R script along with their data. However, I ran into trouble running the R script. Please take acareful look at the following examples I encountered until line 121. I stopped there to follow the script.

Line 12 tries to import a csv file using readxl::read_excel which is for importing Excel files. I recommend to use read.csv instead.

Line 49 uses the column ID_FINAL but it is called id in the csv file as far as I can see. Lines 53-63 perform the renaming of columns. This command should be before line 49.

Lines 111-118:

Error in ncol[, 9] : object of type 'closure' is not subsettable

When changing line 111 to:

for (i in 1:ncol(param[,1:9])) {

I get this error:

Error in do.call(paste, c(min, parameter, max)) :

second argument must be a list

Line 121 loads an R script that I did not find at the Github repo:

source("~/R/functions/min_max_norm.R")

Reviewer #2: Nice to see that my suggestions have been useful. I advice to publish this version. For future research on the same topic, science-society interaction and communication, it would be valuable to not just accept the limitation of the data sources available at your desktop, but to point out possible sources of data in those contexts where the communication actually takes place.

7. PLOS authors have the option to publish the peer review history of their article (what does this mean?). If published, this will include your full peer review and any attached files.

Reviewer #1: No

Reviewer #2: No

---

## [Author Response · Author response to Decision Letter 1]

18 Apr 2022

Please find attached a rebuttal letter.

---

## [Decision Letter · Decision Letter 2]

13 May 2022

A value creation model from science-society interconnections: Archetypal analysis combining publications, survey and altmetric data

PONE-D-21-16293R2

Dear Dr. Robinson-Garcia,

We’re pleased to inform you that your manuscript has been judged scientifically suitable for publication and will be formally accepted for publication once it meets all outstanding technical requirements.

Kind regards,

Lutz Bornmann

Academic Editor

PLOS ONE

Additional Editor Comments (optional):

Reviewers' comments:

Reviewer's Responses to Questions

**Comments to the Author**

1. If the authors have adequately addressed your comments raised in a previous round of review and you feel that this manuscript is now acceptable for publication, you may indicate that here to bypass the “Comments to the Author” section, enter your conflict of interest statement in the “Confidential to Editor” section, and submit your "Accept" recommendation.

Reviewer #1: All comments have been addressed

Reviewer #2: All comments have been addressed

2. Is the manuscript technically sound, and do the data support the conclusions?

Reviewer #1: Yes

Reviewer #2: Yes

3. Has the statistical analysis been performed appropriately and rigorously? 

Reviewer #1: Yes

Reviewer #2: Yes

4. Have the authors made all data underlying the findings in their manuscript fully available?

Reviewer #1: Yes

Reviewer #2: Yes

5. Is the manuscript presented in an intelligible fashion and written in standard English?

Reviewer #1: Yes

Reviewer #2: Yes

6. Review Comments to the Author

Reviewer #1: I think that the documentation via RPubs is a very good idea. However, when I knit the file notebook.Rmd an error occurs. I have to change line 50:

old line 50:

ï..id,

new line 50:

id,

After this change, the file works fine for me.

Reviewer #2: This last revision can be published as it is. All my comments have been responded to in a satisfying way.

7. PLOS authors have the option to publish the peer review history of their article (what does this mean?). If published, this will include your full peer review and any attached files.

Reviewer #1: No

Reviewer #2: **Yes: **Gunnar Sivertsen

---

## [Editor Report · Acceptance letter]

24 May 2022

PONE-D-21-16293R2 

A value creation model from science-society interconnections: Archetypal analysis combining publications, survey and altmetric data 

Dear Dr. Robinson-Garcia:

I'm pleased to inform you that your manuscript has been deemed suitable for publication in PLOS ONE. Congratulations! Your manuscript is now with our production department. 

Kind regards, 

on behalf of

Dr. Lutz Bornmann 

Academic Editor

PLOS ONE